# Lyoprotective Effects of Mannitol and Lactose Compared to Sucrose and Trehalose: Sildenafil Citrate Liposomes as a Case Study

**DOI:** 10.3390/pharmaceutics13081164

**Published:** 2021-07-28

**Authors:** María José de Jesús Valle, Andreía Alves, Paula Coutinho, Maximiano Prata Ribeiro, Cristina Maderuelo, Amparo Sánchez Navarro

**Affiliations:** 1Department of Pharmaceutical Sciences, Faculty of Pharmacy, University of Salamanca, 37007 Salamanca, Spain; mariajosedj@usal.es (M.J.d.J.V.); cmaderuelo@usal.es (C.M.); 2Institute of Biomedical Research of the University of Salamanca (IBSAL), 37007 Salamanca, Spain; 3CPIRN-IPG—Center of Potential and Innovation of Natural Resources, Polytechnic Institute of Guarda, 6300-559 Guarda, Portugal; andreia_alves_23@hotmail.com (A.A.); coutinho@ipg.pt (P.C.); mribeiro@ipg.pt (M.P.R.); 4CICS-UBI—Health Sciences Research Centre, University of Beira Interior, 6200-506 Covilha, Portugal

**Keywords:** liposomes, lyophilization, lyoprotectants, lactose, sildenafil citrate, pulmonary drug delivery

## Abstract

The lyoprotective effects of mannitol and lactose have been evaluated in the production of sildenafil citrate liposomes. Liposomes were prepared by mixing the components under ultrasonic agitation, followed by a transmembrane pH gradient for remote drug loading. Mannitol and lactose, as compared to sucrose and trehalose, were used as the stabilizing agents, and different freeze-drying cycles were assayed. The remaining moisture and the thermal characteristics of the lyophilized samples were analyzed. Size, entrapment efficiency, biocompatibility, and cell internalization of original and rehydrated liposomes were compared. The type of additive did not affect the biocompatibility or cell internalization, but did influence other liposome attributes, including the thermal characteristics and the remaining moisture of the lyophilized samples. A cut-off of 5% (*w*/*w*) remaining moisture was an indicator of primary drying completion—information useful for scaling up and transfer from laboratory to large-scale production. Lactose increased the glass transition temperature to over 70 °C, producing lyoprotective effects similar to those obtained with sucrose. Based on these results, formulations containing liposomes lyophilized with lactose meet the FDA’s requirements and can be used as a biocompatible and biodegradable vehicle for the pulmonary delivery of therapeutic doses of sildenafil citrate.

## 1. Introduction

Among colloidal carriers, liposomes are safe for pulmonary administration in humans and animals [1,2,3,4], and much progress has been made in the formulation of liposomal drugs. Nevertheless, liposomal instability remains one of the main limitations of the translational development of liposome-based formulations [5]. The lyophilization of liposomes is considered the best approach for avoiding drug leaking and vesicle fusion/aggregation, as well as for protecting the lipid bilayer from oxidation [6,7,8,9]. Over the decades, much work has been carried out in this field and, as a result, different lyoprotectants have been proposed as stabilizing agents, with sugars being considered the best option [5,10]. The water replacement and vitrification hypotheses have been proposed to explain liposome cryo- and/or lyoprotection [11,12], and the refinement of these theories has been explored [13]. The literature provides information about the selection of lyoprotectants according to liposome composition and drug loading [14,15,16], and the effects of disaccharides in the gene silencing activity of siRNA lipoplexes have also been assessed [17]. In addition, studies on the administration route for liposomal formulations have been carried out in relation to parenteral drug development [5,18], but not for drug inhalation. This is, however, an important issue, because the regulatory requirements for the marketing of pharmaceutical products are specific to each administration route, and the additives are particularly restricted for inhaled formulations. In respiratory pathologies, the lungs are the target, and in these cases pulmonary administration can provide an advantageous alternative to systemic drug delivery. On one hand, the pharmacokinetic variability affecting pulmonary drug uptake is avoided and a fixed amount of the drug is delivered at the diseased site. On the other hand, the side effects associated with systemic drug exposure are minimized due to lower levels of the drug in blood as compared to oral or parenteral administration. Passive lung targeting by focal drug delivery is currently a priority for several drugs, a topic that has drawn much attention in recent reviews [19,20], including for sildenafil citrate (SC), a 5-phosphodiesterase (PDE-5) inhibitor drug approved in 2005 for the treatment of pulmonary arterial hypertension [21,22]. Solid oral formulations as well as parenteral injection of this drug have been approved for adults and children by the FDA and the European Medicines Agency. Although definitive guidelines on dosage regimens have not yet been established, conservative doses are currently recommended given the serious side effects observed with high doses [23]. Some studies have focused on sildenafil formulations based on polymeric and lipid solid nanoparticles for sustained pulmonary delivery [24,25,26,27,28,29,30], but liposome-based formulations have not been proposed yet. According to the above considerations, the aim of this work was to investigate the stabilizing effects of additives approved for pulmonary drug administration (mannitol and lactose) when used for the lyophilization of liposomes loaded with SC. For this purpose, drug-loaded liposomes were prepared using an eco-friendly procedure, and different freeze-drying cycles using mannitol, lactose, sucrose and trehalose were assayed. The size, entrapment efficiency, biocompatibility, and cell internalization of liposomes before and after lyophilization were determined, and the remaining moisture content and thermal characteristics of the lyophilized samples were analyzed.

## 2. Materials and Methods

### 2.1. Materials

Egg L-α-phosphatidylcholine (EPC), lanolin cholesterol (Ch), and formic acid were purchased from Merck KGaA, Darmstadt, Germany. Sildenafil citrate (SC) was obtained from Acofarma, Madrid, Spain. Mannitol, lactose, and sucrose were from Guinama, Valencia, Spain. Trehalose was purchased from AppliChem GmbH, Darmstadt, Germany. Coumarin-6 was obtained from Acros Organics, Geel, Belgium. Acetonitrile HPLC reagent was purchased from Fisher Scientific, Madrid, Spain. Ultra-pure water was obtained using a Milli-Q A10 system from Merck Millipore, Darmstadt, Germany. The Chromafil^®^ PET-45/25 and PET-20/25 (0.45 and 0.20 µm) syringe filters were purchased from Nacherey-Nagel, Panreac Química S.L.U., Barcelona, Spain. Karl Fischer Aqualine Complete 5 and dry methanol were purchased from Fisher Scientific, Madrid, Spain.

Normal human dermal fibroblasts (NHDF) were purchased from PromoCell, Labclinics, S.A., Barcelona, Spain. Dulbecco’s modified Eagle’s medium/nutrient mixture F-12 Ham (DMEM)-F12, paraformaldehyde (PFA), and phosphate-buffered saline solution (PBS) were purchased from Sigma-Aldrich, Sintra, Portugal. L(+)-ascorbic acid was purchased from Panreac Química S.L.U., Barcelona, Spain. Sodium hydroxide (NaOH) was purchased from José Manuel Gomes dos Santos, Lda. Odivelas, Portugal. Fetal bovine serum (FBS) and bovine serum albumin (BSA) were purchased from Biowest, Nuaillé, France. Glucose was purchased from Labchem, Santo Antão do Tojal, Portugal.

Penicillin/streptomycin amphotericin B (100×) was purchased from Lonza, Barcelona, Spain; trypsin was purchased from SAFC Biosciences^TM^, St. Louis, MO, USA. Ethylenediamine tetraacetic acid (EDTA) was purchased from Guinama Valencia, Spain. MTT (thiazolyl blue tetrazolium bromide), dimethyl sulfoxide (DMSO), and ethanol absolute were purchased from VWR, Barcelona, Spain. Hoestcht 33342 and wheat germ agglutinin (WGA) were purchased from ThermoFisher, Carlsbad, CA, USA.

### 2.2. Liposomes

#### 2.2.1. Preparation

Liposomes with and without additive were prepared by mixing the components under ultrasonic agitation, followed by the application of a transmembrane pH gradient for remote drug loading [31]. For liposomes without additive, EPC and Ch (molar ratio 5.5/1) were mixed with citrate buffer solution at pH = 3.2 (blank liposomes, B) or 1 mg/mL SC buffer solution (SC-loaded liposomes) to a lipid concentration of 1.8% *w*/*v*. The mixtures were placed in a Fisher Scientific FB 15061 ultrasonic bath (50 Hz) at 50 ± 2 °C for 30 min. The resulting suspensions were filtered 10 times across syringe filters (Cromafil^®^ PET) of a predetermined pore size (0.45 µm or 0.20 µm). The filtrated samples were kept at room temperature for 60 min and stored at 4 °C for 1 h. A transmembrane pH gradient was then applied for remote drug loading as follows: NaOH 0.1 N was added to the liposome samples until pH = 7.0 ± 0.1 and maintained for 20 h at 4 ± 0.2 °C in a shaking water bath (Unitronic OR Selecta P) to facilitate the drug diffusion and accumulation in the water core. Liposomes with coumarin 6 as the fluorescent probe (C liposomes) were also prepared. In this case, 400 µL of coumarin solution in ethanol (0.01% *w*/*v*) was added to the lipid mixture before ultrasonic agitation, and the rest of the procedure was the same as previously described. Liposomes with additive were prepared following the above procedure, but either mannitol (M), lactose (L), sucrose (S), or trehalose (T) at 4% *w*/*w* was incorporated in the buffer solution.

#### 2.2.2. Characterization

The size (Dh), polydispersity index (PDI), and zeta potential of liposomes, with and without additive, were analyzed via the dynamic light scattering (DLS) technique using a Zetasizer Nano ZS (Malvern Instruments, Malvern, UK). The analysis was performed at 25 °C and a scattering angle of 173° after appropriate dilution (×100 or ×1000) with Milli-Q water to avoid the phenomenon of multiple scattering.

Entrapment efficiency (EE) and drug loading (DL) were determined as follows: liposome suspensions were centrifuged at 14,000 rpm for 45 min at 6 °C to separate the non-entrapped SC. The drug in the supernatant was quantified by HLPC. A Purosphere STAR rpE18 50 cm × 4.0 mm 3 µm column and a mixture of 0.1% formic acid in water and acetonitrile (70/30 *v*/*v*), adjusted to pH 4.2 using triethanolamine, were used at a flow rate of 1.5 mL/min. The UV detector was set at 292 nm (HPLC system with Waters Alliance 2695 separation module, 2998 photodiode array detector and empower processor system). Standard solutions (25–500 µg/mL) were prepared with citrate buffer at pH 3.2 adjusted to pH 7 using 0.1 N NaOH. Standard solutions containing M, L, S or T were also prepared to test the selectivity of the analytical method (results in Appendix A).

The amount of SC in the liposomes (Qin) was estimated as follows:Qin = Qt − Qs(1)
where Qt is the total amount of SC in the sample and Qs the amount of SC in the supernatant.

The entrapment efficiency was estimated using the following equation:EE (%) = (Qin/Qt) × 100(2)

The DL was determined as the Qin/Qlipid ratio, where Qlipid is the amount of lipid in the liposomes.

The kinematic viscosity of the suspensions (η) with and without the corresponding additive was determined using a capillary viscometer by measuring the time it took the sample to flow through the capillary under the influence of gravity at 25 ± 2 °C. The instrument was calibrated with Milli-Q water (ρ = 1 g/mL; η = 1 mm^2^/s) and the viscometer constant (K, mm^2^/s^2^) was estimated. The viscosity of the liposome suspensions was then calculated as follows:η = K × t,(3)
where t is the time it took for the liposome suspension to flow from the lower to the upper mark of the viscometer.

The morphology of liposomes was studied by scanning electron–transmission microscopy (SEM) using a Zeiss EVO HD259 microscope. Diluted samples were fixed with poly-L-lysine and osmium, and acetone was used as the desiccant product.

### 2.3. Biocompatibility and Cell Internalization

#### 2.3.1. Biocompatibility Assessment

Normal human dermal fibroblasts (NHDF) were used to evaluate the effects of liposomes with and without additives on cellular growth. Cells were first seeded in T-flasks of 75 cm^2^ containing 10 mL of DMEM-F12 medium supplemented with heat-inactivated FBS (10% *v*/*v*) and 1% antibiotic/antimycotic solution, and then incubated at 37 °C in a 5% CO_2_ humidified atmosphere. When cells reached confluence, they were released by incubation in 0.18% trypsin (1:250) and 5 mM EDTA. Cell growth was monitored using an Optika inverted light microscope equipped with an Opikam B5 (BG, Bergamo, Italy) digital camera. The MTT assay was then used, following ISO 10993–5, to evaluate cell viability in the presence of liposomes. Cells were seeded in 96-well plates with DMEM-F12, using a cellular density of 2 × 10^4^ cells/well and incubated at 37 °C in a 5% CO_2_ humidified atmosphere for 24 h. After 24 h, the culture medium was removed and substituted with fresh culture medium mixed with sterile liposome suspension diluted to 1:100, which corresponds to 0.018% *w*/*v*. This concentration is higher than that reached during lung exposure, as estimated from the amount of liposomes that provide a suitable SC pulmonary dose. The mixture was maintained at 37 °C in a 5% CO_2_ humidified atmosphere for 24 h (*n* = 5). After 24 h the culture medium was removed and replaced with 50 µL of MTT. The cells were then incubated for 3 h at a 37 °C in a 5% CO_2_ humidified atmosphere. Thereafter, MTT was removed and 100 µL of dimethyl sulfoxide (DMSO) was added and gently agitated for 15 min at room temperature to dissolve the formazan crystals. The absorbance was then registered at 570 nm using a microplate reader (Multiskan GO—Thermo Scientific). Subsequently, 96% ethanol was added to the cells used as positive controls (K^+^) (dead cells), whereas untreated cells (without liposomes) were used as negative controls (K^−^). The mean cell viability values have been expressed as the percentage of K^−^.

#### 2.3.2. Cell Internalization

Confocal laser scanning microscopy (CLSM) was used to study the cell internalization of liposomes. Cells were seeded (4 × 10^4^ cells/well) on glass-bottomed coverslips and incubated for 24 h at 37 °C in a 5% CO_2_ humidified atmosphere [32]. After 24 h, the culture medium was removed and substituted with fresh medium mixed with sterile suspensions of liposomes prepared with C as a fluorescent probe at 37 °C in a 5% CO_2_ humidified atmosphere. Cells without liposomes were used as control. After 12 h of incubation, the samples were taken for labeling preparation. The fibroblast nucleus was labelled with Hoestcht 33342 (5 µg/mL) and the cytoplasm with WGA (5 µg /mL) then fixed with PFA (4% *v*/*v*) in PBS for 10 min at room temperature. Imaging analyses were performed with a Zeiss LSM710 laser scanning confocal microscope (Carl Zeiss AG, Jena, Germany).

### 2.4. Lyophilization

A total of 4% (*w*/*w*) M, L, S, or T was added to the liposome suspension (additive/EPC mass ratio = 2.6) before freezing, and the mixtures were softly agitated for 3 min. Glass vials were filled with 1.0 mL of mixture and frozen at −80 ± 3 °C (Nuaire Ultralow Freezer). Annealing to induce the crystallization of the lyoprotectant agents was assayed. Since experimental values of T’g were not determined, information from the literature was used, and annealing was performed at −33 ± 2 °C for 4 h (Whirlpool AFG 639, Whirlpool, Benton Harbor, MI, USA). A laboratory freeze-drying instrument (Telstar Cryodos, Telstar S.L., Terrassa, Spain), with a condenser temperature of −80 ± 4 °C and chamber pressure of 0.008 ± 0.002 mBar (Pirani gauge), was used. The temperature during primary drying was −45 ± 5 °C. Batches of 30–35 vials were employed for each freeze-drying cycle. Protocols with and without annealing, and with and without secondary drying (10–20 °C), were carried out with the same chamber pressure as was used in the primary drying.

### 2.5. Moisture Content

The remaining moisture (RM) in each freeze-dried cake was determined using the Karl Fisher method, transferring 0.1 g of sample to the titration cell. The mass of the analyzed sample was determined by analytical balance (Mettler Toledo XS105DU, Mettler Toledo, Greifensee, Switzerland) and the volumetric water content was measured using a Metrohm 870 KF Titrino plus KF titrator. The results are shown as the dry product RM *w*/*w* value.

### 2.6. Differential Scanning Calorimetry (DSC)

The thermal analyses of fresh and lyophilized samples were carried out by DSC. The gel-to-liquid crystalline phase transitions (Tm) of dry EPC (control), fully hydrated EPC (fresh liposomes), and freeze-dried EPC (lyophilized cakes) were determined. In addition, the glass transition temperatures (Tg) of the products lyophilized with M, L, S, or T were analyzed. Experiments were performed using a Mettler Toledo DSC-1. The samples (1.0 mg) were transferred to 40 μL aluminum pans, sealed, and scanned from −20 to 150 °C at 10 °C/min. The analyses of the thermograms and the transition temperatures were performed using the STARe software package. Tm was recorded as the peak temperature of the endotherm in the lipid gel-to-liquid crystalline phase transition and Tg as the onset temperature of the endotherm for glass transition, both during the heating scan [33].

### 2.7. Reconstitution

After a visual inspection and revision of cake appearance, 1 mL of ultra-pure water was added to the dry cake during manual shaking. The time required for the complete disappearance of solid content was determined by visual inspection and recorded as the reconstitution time. The EE, DL, Dh, PDI, and zeta potential in the rehydrated samples were determined as described for the original liposomes.

### 2.8. Statistical Analysis

Comparison of results was performed using one-way analysis of variance (ANOVA) with Tukey´s post hoc test. Statistical significance was considered at *p*-values ≤ 0.05.

## 3. Results

### 3.1. Fresh Liposomes

#### 3.1.1. Liposomes without Additives

The application of a transmembrane pH gradient to liposomes without any additive led to an EE% > 60%. In addition, the size of liposomes affected the EE%, with mean values of 83.10 ± 9.18% and 62.10 ± 9.32% for the larger and smaller liposomes, respectively. Filtration through a 0.20 µm filter produced liposomes with mean Dh and PDI values of 162.73 ± 15.01 nm and 0.17 ± 0.03, respectively, for B liposomes, and 175.40 ± 1.32 nm and 0.13 ± 0.02, respectively, for SC liposomes. The differences in Dh and PDI between B and SC liposomes did not show statistical significance (*p* = 0.11 and *p* = 0.18, respectively). Filtration through a 0.45 µm filter produced larger liposomes showing higher PDI values (Dh = 270.71 ± 6.70 nm and PDI = 0.28 ± 0.06 for B liposomes and Dh = 293.34 ± 13.69 nm and PDI = 0.32 ± 0.08 for SC liposomes), with statistically significant differences between B and SC liposomes for Dh (*p* < 0.05). Negative zeta potential values in the range of −11.23 ± 1.39 mV to −9.20 ± 1.11 mV were obtained for all liposomes without additives, regardless of cargo and size.

#### 3.1.2. Liposomes with Additive

For liposomes containing an additive, the EE% did not reach the values mentioned above. However, lower EE% values were obtained for both large and small liposomes, and the small liposomes with T presented the lowest EE% values (44.92 ± 7.10%). The additives also affected the Dh and PDI of small and large liposomes. A statistically significant increase in Dh was observed when SC liposomes without additive (SC) were compared to those containing mannitol or lactose (*p* < 0.05). In terms of PDI, differences between SC, SC + M, SC + L, and SC + S were found (*p* < 0.05). The zeta potentials of small and large liposomes were not affected by the additives, and all liposomes showed values in the range of −11.23 ± 1.39 mV to −7.84 ± 1.12 mV. In addition, higher ƞ values were found for liposome samples containing an additive when compared to those without additives, and the differences were statistically significant (*p* < 0.05). Among the additives, the differences in ƞ were not statistically significant (*p* = 0.81).

Table 1 and Figure 1 summarize the results obtained for fresh liposomes prepared with and without additives.

### 3.2. Biocompatibility and Cell Internalization

The microscopic images (Figure 2) revealed that cells adhered and proliferated in the presence of liposomes, regardless of the size and cargo, and they presented phylopodia, as observed in the negative control (K^−^).

In the positive control (K+), with cell death induced by the addition of ethanol, neither adhesion nor proliferation was observed; however, a spherical shape typical of dead cells was observed. The results of the MTT assay (Figure 3) show that B and SC liposome samples, as well as SC + L and SC + S, did not produce cytotoxic effects. After confirming that the Dh, PDI, and zeta potential values of C liposomes were not statistically different from those of SC liposomes, a cell uptake and internalization assay was performed on C liposomes with and without additive.

Figure 4 shows the confocal laser microscopy images. The dot-like staining pattern was localized adjacent to the cell nuclei, while the plasma membrane remained unstained, suggesting the nonspecific binding of liposomes to the membrane. These results prove that the liposomes prepared as previously described have the ability to deliver a drug into cultured human cells, irrespective of their size and the additive used.

### 3.3. Lyophilization

Table 2 shows the effect of annealing on remaining moisture. According to our results, no sample benefited from annealing. Moreover, statistically higher values of RM were observed for M samples when annealing was performed (3.45 ± 0.64 versus 2.74 ± 0.77% for small liposomes and 3.93 ± 0.43 versus 3.36 ± 0.62% for large liposomes). These results agree with previous findings that annealing at −33 ± 2 °C induces the crystallization of M, the formation of a dense top layer, and an increase in resistance to vapor flow during primary drying [34,35,36,37]. For L and S, a slight decrease in RM was observed when annealing was performed, but the differences were not statistically significant (*p* = 0.07).

The results of RM show that annealing did not improve water removal during primary drying. Since the crystallization of lyoprotectants was not essential for our samples due to their low solid content, annealing at −33 °C [35,36,37] was not performed in the rest of experiments. Primary drying completion was initially tested by trial and error, and it was observed that a drying time below 18 h led to collapsed/melted cakes. The extension of primary dying time caused a progressive reduction in the occurrence of collapse, and after a period of 24 h, collapsed samples were not observed. Analysis of the water content in the samples revealed that collapsed cakes had an RM ≥ 6%, regardless of the liposome size, the additive used, or the temperature of secondary drying (10–20 °C range), while uniform cakes were observed for samples with an RM ≤ 5%. The elimination of unfrozen water during secondary drying was very slow for all samples, particularly when this was carried out at under 20 °C. Accordingly, an RM of 5% was the starting point for secondary drying, and 20 °C was the temperature selected for this phase. Table 3 shows the RM of the cakes and the characteristics of the rehydrated liposomes in samples lyophilized under the conditions finally selected (freezing at −80 °C; 24 h of primary drying and 20 h of secondary drying at 20 °C).

Under these conditions, secondary drying was faster for samples with M than for those containing L, S, or T. A reduction in RM of 49.13% was achieved for the liposome sample with M, while reductions of 21.42%, 27.02%, and 13.55% were observed for L, S, and T, respectively. Cakes containing small liposomes showed slightly higher RM values compared to those with large liposomes, but the differences were not statistically significant (*p* > 0.05). Scanning electron microscopy (SEM) images of the lyophilized cakes (Figure 5) reveal a smooth surface and no evidence of the crystal edges that would be beneficial to the stability of the samples.

### 3.4. Reconstitution

Lyophilization under the above conditions produced cakes of uniform appearance (Appendix A) with short reconstitution times (under 30 s), regardless of the liposome size or the additive used. In any case, the Dh and PDI values of the rehydrated liposomes were higher than the original values. This increase was less significant for samples containing L or S than for samples containing M or T, but differences between the original and rehydrated liposomes were statistically significant in all cases (*p* < 0.05). With respect to the EE, statistical differences between the original and rehydrated liposomes were not observed for the large liposomes (*p* = 0.37). For the small liposomes, however, the EE increased from 62.10 ± 9.32 (originals) to 97.96 ± 0.33%, 85.10 ± 3.10%, 82.56 ± 1.71%, and 84.91 ± 2.76% for the rehydrated samples containing M, L, S, and T, respectively. The SEM images of fresh and rehydrated liposomes were compared (Figure 6). Differences in morphology were not apparent, and a spherical shape was observed in the original and rehydrated liposomes.

### 3.5. Differential Scanning Calorimetry

The thermal characteristics of EPC in fresh and lyophilized liposomes were analyzed via the DSC thermograms (Appendix A). Figure 7 illustrates the results. The Tm (left panel) appeared at −13 °C for the fully hydrated lipid (EPC in fresh liposomes) and 29 °C for the dry EPC; these values agree with data in the literature [12]. The thermograms from cakes with L or S showed a single peak at −13 °C, as was found for fully hydrated EPC. For the M and T samples, however, an additional endotherm peak at 18–21 °C was observed. These data prove that L and S (additive/EPC mass ratio of 2.6) can act as efficient lyoprotectants since both reduced the Tm to the value of fully hydrated EPC. By contrast, M and T only caused a slight displacement of the endotherm peak from 29 to 18–21 °C. With respect to Tg, for L and S, the lower the RM%, the higher the Tg, and temperatures of 78.92 and 68.64 °C were reached for L (RM = 2.14%) and S (RM = 1.95%), respectively. A linear relationship between the RM% and the Tg values was found in both cases (Appendix A). For M and T, however, a linear relationship between RM% and Tg was not found. Notice that the onset was not clear in DSC thermograms for trehalose, and therefore the Tg values were not accurately estimated for this lyoprotectant.

## 4. Discussion

Liposomes of different sizes and compositions were obtained using an eco-friendly procedure of mixing the components under controlled ultrasonic agitation. The application of a transmembrane pH gradient produced liposomes with EE% values that depended on the size and composition of pre-formed liposomes. The DL estimated from the corresponding EE% reveals that the large liposomes without additive showed the highest DL value (49.86 ± 5.52 mg SC/g of lipid), while the small liposomes with T showed the lowest DL (26.94 ± 4.26 mg SC/g of lipid). Diffusion through the lipid bilayer is the underlying mechanism for remote loading when a transmembrane pH gradient is applied. The lower EE values registered for samples containing L, S, or T can be attributed to the higher viscosity of these samples, but the increase in ƞ did not affect the samples containing M. Therefore, the specific interaction between the phospholipid head groups and the sugar molecules is likely the mechanism hindering drug diffusion through the lipid bilayer for SC liposomes prepared with L, S, and T. The size and polydispersity of the liposomes obtained via the procedure described were indeed affected by the additives; however, the negative zeta potential was not affected, and the values were in the range of −11.23 ± 1.39 mV to −7.84 ± 1.12 mV. Based on the above, the SC liposomes obtained in this study showed the following critical attributes: (A) biocompatibility and biodegradability, since EPC and Ch are both components of the pulmonary surfactant; (B) no risk of solvent residuals, since liposomes were prepared in the absence of organic solvents; (C) the size and zeta potential accorded with those values recommended for particles used for pulmonary administration; (D) the DL values were high enough to enable a level of pulmonary drug exposure higher than that achieved with the current administration routes used for pulmonary hypertension. Moreover, the results from the biocompatibility and cell internalization experiments show that liposomes with or without additive did not produce cytotoxic effects and were able to enter the cells. Accordingly, these liposomes meet the regulatory requirements for pulmonary formulations and seem to be suitable for the inhalation of therapeutic SC doses.

As previously mentioned, liposomal instability remains a major translational limitation in the development of liposome-based formulations. Due to the regulatory restrictions on additives used for pulmonary delivery, it was necessary to analyze the lyoprotective effects of mannitol and lactose considering that lyophilization is proposed for its stabilization. SC was selected as the prototype drug as there is interest in improving the benefit/risk ratio of this drug for its use in the clinical treatment of pulmonary hypertension. The RM, Tm, and Tg of the lyophilized products were selected as critical attributes because they are directly related to physical and chemical stability [13]. Our results confirm that annealing does not contribute to improving the drying rate, and this was accordingly removed from the final lyophilization protocol. Secondary drying rates were found to be rather slow for all formulations, even for those containing M, which slightly improved the drying rate as compared to the rest of the additives tested. An interesting result was the determination of an RM cut-off value of 5%, which would allow the safe secondary drying of the studied formulations. To our knowledge, RM cut-off values have not previously been proposed as an indicator of primary drying completion for liposomal formulations. This value is specific to the formulations tested here, and facilitates scaling up to any lyophilization equipment.

The comparison of the additives revealed that M is beneficial to drying, but not to stabilizing, the liposomes. In addition, the thermograms proved that L and S are able to reduce the Tm to the value associated with fully hydrated EPC, while M and T only produced a slight change. Tm reduction enables the liposomal phospholipid membrane to retain a single phase during both drying and rehydration, avoiding structural transitions and drug leakage [38]. Tg has been related to the stabilization of liposomes in the glass matrix. After freeze-drying, amorphous components remain vitrified only in cases in which the storage temperature is well below the Tg of the glass [16]. As a rule of thumb, a difference of 50 °C between the storage temperature and the Tg has been proposed for sample preservation. In the case of the L samples, RM values < 2.2% led to Tg values > 70 °C. Accordingly, L cakes can be stored at room temperature as long as RM ≤ 2%. With respect to the EE%, the differences found in the small, rehydrated liposomes compared to the originals may be explained by the amphiphilic nature of SC with the predominance of cationic, anionic, or neutral species depending on the pH. Some studies have shown that the trans-membrane flux of SC is greatest at pHs ≥ 10 due to the predominance of the anionic species with the highest permeability [18,39]. Our hypothesis was that cryo-concentration would be beneficial to the permeation of SC across the lipid bilayer by increasing the pH, the anionic species, and the concentration gradient, resulting in rehydrated liposomes with a higher EE%. This effect was not observed for large liposomes due to the small quantity of non-entrapped drug in the original samples. Similar results have been found for salmon calcitonin liposomes [40]. The increases in Dh and PDI values observed in rehydrated liposomes from samples containing L, S, or T do not affect their suitability for pulmonary drug delivery. Liposomes with Dh values up to 813.00 ± 9.21 nm have been found suitable for pulmonary drug delivery; despite the higher polydispersity of rehydrated liposomes, their Dh range is narrower than that above [41]. Lyophilized SC liposomes using L exhibited the following critical attributes: (A) a reduction in Tm to the value associated with fully hydrated EPC, which avoids structural transitions and drug leakage; (B) a Tg > 70 °C, which stabilizes liposomes in the glass matrix and allows for storage at room temperature; (C) a DL ≥ 50 mg SC/g of lipid, which is high enough for the pulmonary administration of therapeutic doses; (D) Dh values of rehydrated liposomes in agreement with the reported sizes of particles suitable for pulmonary administration. Taken together, the results of this study indicate that SC liposomes lyophilized with lactose produce a stabilized liposomal formulation that fulfils FDA requirements and allows for the pulmonary delivery of therapeutic doses of this drug.

## 5. Conclusions

Biocompatible and biodegradable liposomes with high DL values were prepared using a solvent-free procedure. In vitro assays showed that the liposomes had no cytotoxic effects and were able to enter the cells regardless of size and drug loading. In addition, these properties were not affected by the additives used. With regard to lyophilization, mannitol was useful as a bulking and drying agent, but not as a lyoprotectant. Lactose succeeded in producing cakes with Tm and Tg values associated with stability at room temperature and the avoidance of structural transitions. An RM cut-off value of 5% was found to be an indicator of primary drying completion, which would facilitate the scale-up and transfer from laboratory to large-scale production. Long-term stability and in vivo studies are necessary to confirm the potential of liposomes lyophilized with lactose as being suitable for the pulmonary delivery of SC to clinically treat pulmonary hypertension. Other drugs used for pulmonary diseases may benefit from this promising liposomal formulation.

## Figures and Tables

**Figure 1 pharmaceutics-13-01164-f001:**
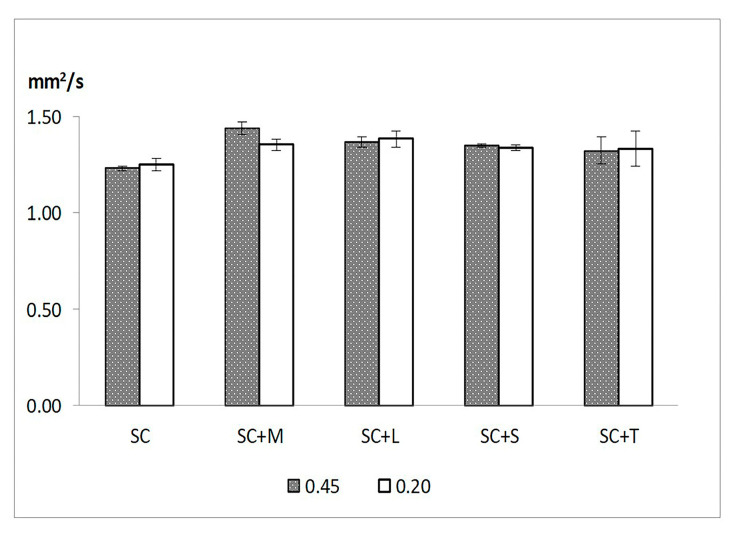
Influence of lyoprotectants on the viscosity of liposome suspensions (SC: sildenafil citrate liposomes without additive; SC + M: sildenafil citrate liposomes with mannitol; SC + L: sildenafil citrate liposomes with lactose; SC + S: sildenafil citrate liposomes with sucrose; SC + T: sildenafil citrate liposomes with trehalose).

**Figure 2 pharmaceutics-13-01164-f002:**
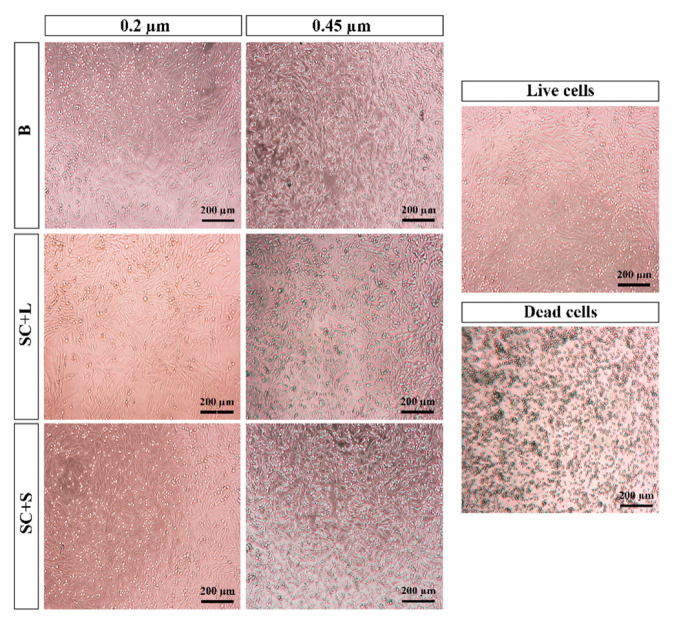
Microscopic images of cell cultures in the presence of liposomes (B: blank liposomes without additive; SC: sildenafil citrate liposomes without additive; SC + L: sildenafil citrate liposomes with lactose; SC + S: sildenafil citrate liposomes with sucrose).

**Figure 3 pharmaceutics-13-01164-f003:**
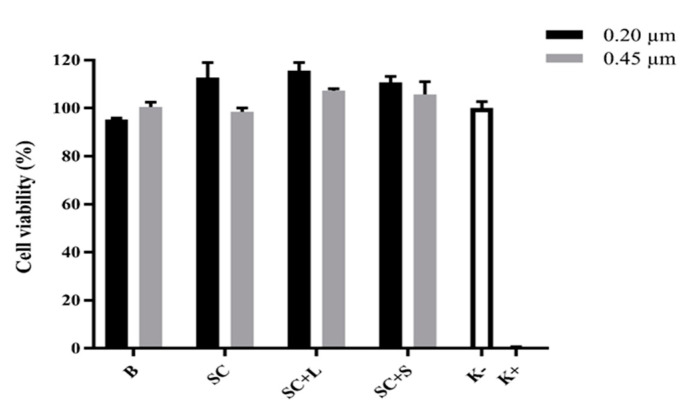
Results from biocompatibility assay performed with blank and drug-loaded liposomes without additives (B and SC, respectively) and sildenafil citrate liposomes with lactose or sucrose (SC + L and SC + S, respectively).

**Figure 4 pharmaceutics-13-01164-f004:**
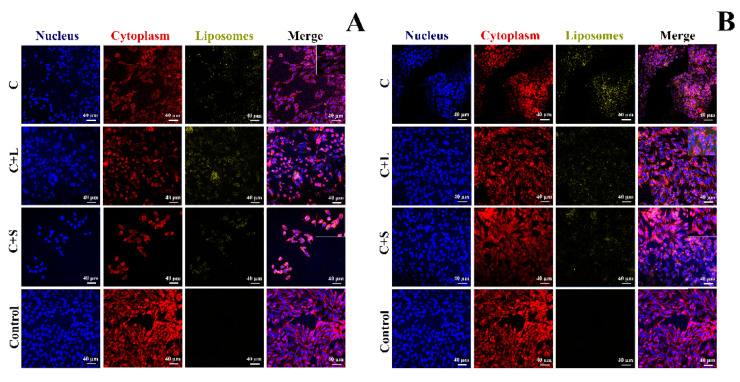
Confocal laser microscopy images obtained for coumarin liposomes without additive (C) and coumarin liposomes with lactose or sucrose (C + L or C + S, respectively), filtered through 0.20 µm (**A** panel) or 0.45 µm filters (**B** panel). Control: cells without liposomes.

**Figure 5 pharmaceutics-13-01164-f005:**
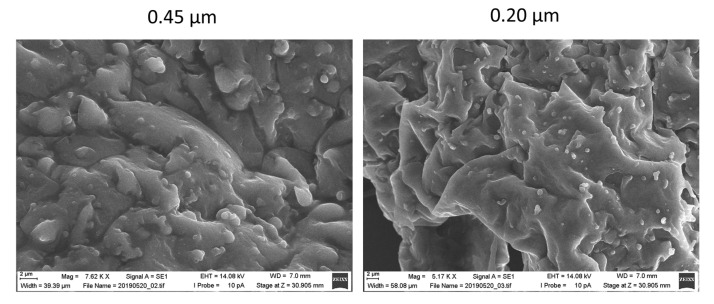
SEM images of lyophilized cakes from lactose samples filtered through 0.20 and 0.45 µm filters.

**Figure 6 pharmaceutics-13-01164-f006:**
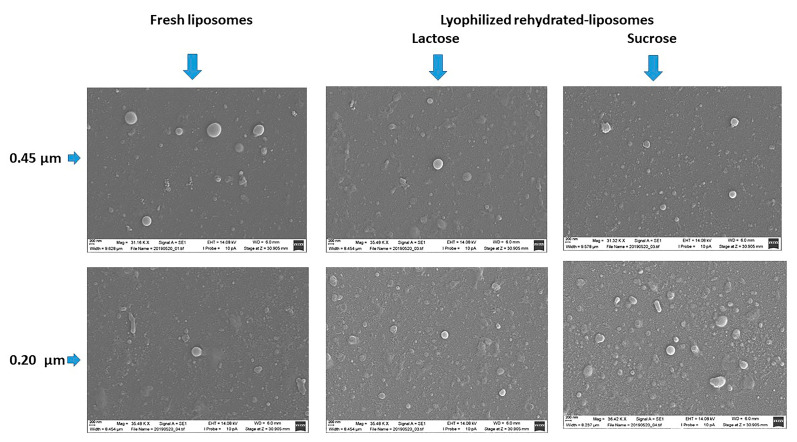
SEM images of fresh and rehydrated liposomes.

**Figure 7 pharmaceutics-13-01164-f007:**
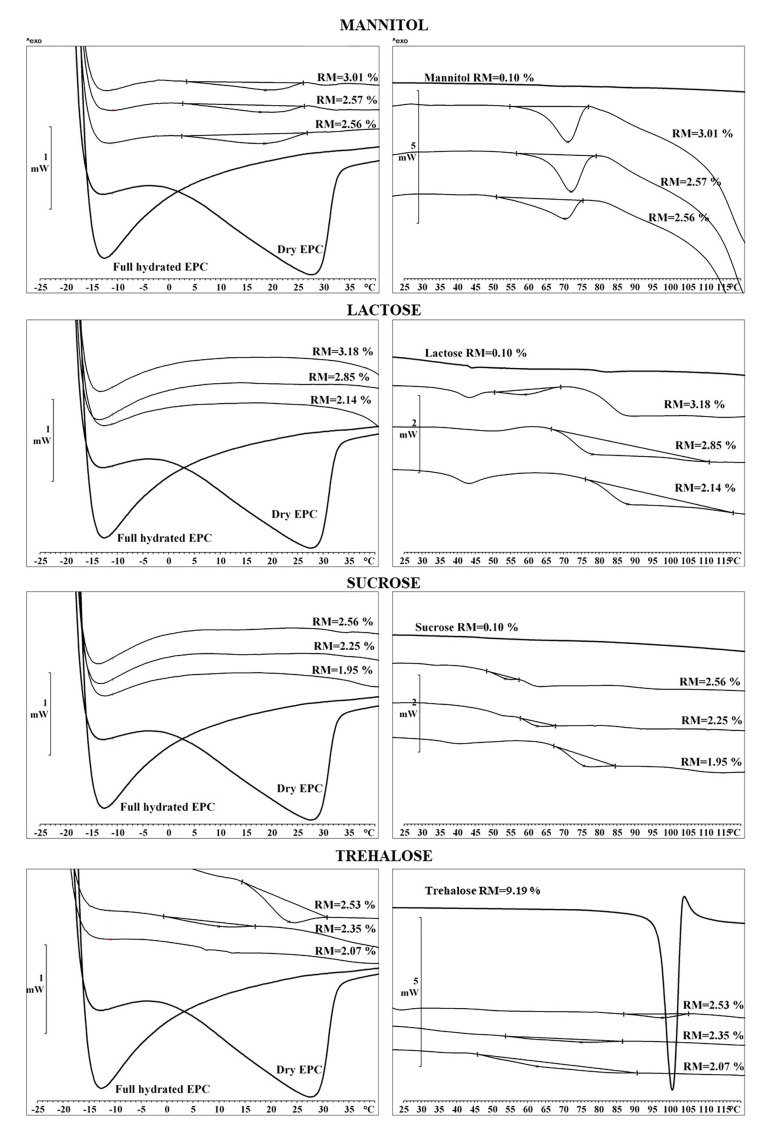
DSC thermograms of dry EPC, full hydrated EPC (fresh liposomes) and EPC in lyophilized liposomes. Influence of additives and remaining moisture (RM) on gel-to-liquid crystalline phase transition (left panel) and on glass transition temperature (right panel).

**Table 1 pharmaceutics-13-01164-t001:** Influence of additives on the characteristics of fresh liposomes. EE, Dh, PDI, and Zeta: entrapment efficiency, size, polydispersity index, and zeta potential, respectively.

		B	SC	SC + M	SC + L	SC + S	SC + T
0.45 µm	EE%	-	83.10 ± 9.18	83.79 ± 1.30	78.30 ± 5.09	65.91 ± 7.40	66.33 ± 10.38
	Dh (nm)	270.71 ± 6.70	293.34 ± 13.69	304.80 ± 7.09	266.60 ± 2.08	272.41 ± 7.00	260.78 ± 27.80
	PDI	0.28 ± 0.06	0.32 ± 0.08	0.27 ± 0.02	0.30 ± 0.04	0.37 ± 0.08	0.23 ± 0.03
	Zeta (mV)	−10.23 ± 1.50	−11.23 ± 1.39	−9.80 ± 0.60	−7.84 ± 1.12	−8.70 ± 1.21	−11.03 ± 0.91
0.20 µm	EE%	-	62.10 ± 9.32	62.31 ± 1.62	49.90 ± 5.40	49.33 ± 6.30	44.92 ± 7.10
	Dh (nm)	162.73 ± 15.01	175.40 ± 1.32	195.83 ± 8.91	205.40 ± 11.40	190.51 ± 17.44	177.82 ± 13.61
	PDI	0.17 ± 0.03	0.13 ± 0.02	0.21 ± 0.08	0.22 ± 0.08	0.20 ± 0.02	0.14 ± 0.04
	Zeta (mV)	−9.20 ± 1.11	−10.62 ± 1.43	−8.72 ± 0.71	−11.01 ± 1.20	−9.30 ± 1.82	−9.44 ± 2.21

B: blank liposomes without additive; SC: sildenafil citrate liposomes without additive; SC + M: sildenafil citrate liposomes with mannitol; SC + L: sildenafil citrate liposomes with lactose; SC + S: sildenafil citrate liposomes with sucrose; SC + T: sildenafil citrate liposomes with trehalose.

**Table 2 pharmaceutics-13-01164-t002:** Influence of annealing on the remaining moisture (% *w*/*w* on a dry product basis) in samples after 24 h of primary drying (SC + M, SC + L, SC + S, and SC + T: sildenafil citrate liposomes with mannitol, lactose, sucrose, and trehalose, respectively).

		SC + M	SC + L	SC + S	SC + T
**Without** **Annealing**	0.45 µm	3.36 ± 0.62	3.57 ± 0.10	3.39 ± 0.41	4.78 ± 0.25
	0.20 µm	2.74 ± 0.77	3.58 ± 0.44	3.54 ± 0.63	4.60 ± 0.36
**With** **Annealing**	0.45 µm	3.93 ± 0.43	3.17 ± 0.39	3.23 ± 0.32	4.89 ± 0.48
	0.20 µm	3.45 ± 0.64	3.25 ± 0.33	3.36 ± 0.42	4.89 ± 0.36

**Table 3 pharmaceutics-13-01164-t003:** Influence of additives on results obtained in the final lyophilization cycle (freezing at −80 °C; 24 h of primary drying and 20 h of secondary drying at 20 °C). RM: remaining moisture in dry cake; EE, Dh, PDI, and Zeta: entrapment efficiency, size, polydispersity index, and zeta potential of rehydrated liposomes, respectively (SC + M, SC + L, SC + S, and SC + T: sildenafil citrate liposomes lyophilized with mannitol, lactose, sucrose, and trehalose, respectively).

		SC + M	SC + L	SC + S	SC + T
Filter 0.45 µm	RM%	1.56 ± 0.24	2.81 ± 0.06	2.58 ± 0.09	2.91 ± 0.2
	EE%	98.20 ± 0.10	84.50 ± 3.19	81.83 ± 5.52	84.64 ± 6.34
	Dh (nm)	757.60 ± 262.19	360.97 ± 84.16	421.50 ± 144.36	412.60 ± 118.24
	PDI	0.57 ± 0.38	0.37 ± 0.13	0.26 ± 0.02	0.35 ± 0.26
	Zeta (mV)	−12.64 ± 3.65	−19.93 ± 5.84	−16.87 ± 3.09	−18.97 ± 6.65
Filter 0.20 µm	RM%	2.92 ± 0.49	3.06 ± 0.09	2.54 ± 0.14	3.01 ± 0.14
	EE%	97.96 ± 0.33	85.10 ± 3.10	82.56 ± 1.71	84.91 ± 2.76
	Dh (nm)	554.87 ± 98.11	344.93 ± 69.22	324.20 ± 12.87	409.13 ± 7.91
	PDI	0.50 ± 0.10	0.45 ± 0.04	0.38 ± 0.10	0.53 ± 0.18
	Zeta (mV)	−17.73 ± 4.20	−16.47 ± 6.80	−17.93 ± 7.93	−19.43 ± 3.55

## Data Availability

The data presented in this study are available within the article.

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
