# Peer review of "Lyoprotective Effects of Mannitol and Lactose Compared to Sucrose and Trehalose: Sildenafil Citrate Liposomes as a Case Study"

_pharmaceutics, 2021, doi:10.3390/pharmaceutics13081164_

Round 1

Reviewer 1 Report

In the paper “Lyoprotective effects of mannitol and lactose compared to sucrose and trehalose. Sildenafil citrate liposomes, as a case study” the authors synthesize and characterize, before and after lyophilization, liposomes with different additives with the purpose of increase stability. The most relevant advantages identified are related to drying rate. Other differences, such as increase in size and PDI, are not clearly commented and justified. The following clarification are recommended to strengthen the paper significance.

  • All formulations showed major increase in size and PDI after lyophilization, the impact of these changes on particles properties should be better described and commented.
  • Biocompatibility and cell internalization were only tested on fresh liposomes and without specifying the concentration of liposomes used. Please include liposomal concentration and higher magnification microscope images. The current images have poor resolution and are not clear enough to confirm any reported result. In particular, the scalebar of the SEM images cannot be red, if is the same across all images the liposomes seem to have all the same size, while the table states otherwise. Please clarify.
  • To better highlight the advantages of the additive it would be nice to include also the characterization of liposomes without additives after lyophilization. Changes in biocompatibility, size, PDI and drying rate should be reported and compared with the ones of liposomes with additives.

Author Response

REVIEWER 1

Comments and Suggestions for Authors

In the paper “Lyoprotective effects of mannitol and lactose compared to sucrose and trehalose. Sildenafil citrate liposomes, as a case study” the authors synthesize and characterize, before and after lyophilization, liposomes with different additives with the purpose of increase stability. The most relevant advantages identified are related to drying rate. Other differences, such as increase in size and PDI, are not clearly commented and justified. The following clarification are recommended to strengthen the paper significance.

We appreciate very much the reviewer´s comments and suggestions and we have made the corresponding changes in the revised version

  • All formulations showed major increase in size and PDI after lyophilization, the impact of these changes on particles properties should be better described and commented.

Answer: the change of size was commented in the fisrt version (section 4, last paragraph): “The increase of Dh values observed in rehydrated liposomes from samples containing L, S or T does not affect their suitability for pulmonary drug delivery, since liposomes with Dh values up to 813.00±9.21 nm have been found suitable for pulmonary drug delivery.”  Following the indication of the reviewer, this pargraph has been extended in the revised version as follows: “The increase of Dh and PDI values observed in rehydrated liposomes from samples containing L, S or T does not affect their suitability for pulmonary drug delivery. Liposomes with Dh values up to 813.00±9.21 nm have been found suitable for pulmonary drug delivery. Despite higher polydipersity of rehydrated liposomes, the Dh range is under the above value. Lines 466-470

  • Biocompatibility and cell internalization were only tested on fresh liposomes and without specifying the concentration of liposomes used. Please include liposomal concentration and higher magnification microscope images. The current images have poor resolution and are not clear enough to confirm any reported result. In particular, the scalebar of the SEM images cannot be red, if is the same across all images the liposomes seem to have all the same size, while the table states otherwise. Please clarify.

Answer: in the first version of the ms the dilution is indicated (section 2.3.1.:”After 24 h the culture medium was removed and substituted by fresh culture medium mixed with sterile liposome suspension with a dilution of 1:100…”.  

1:100 dilution produces a concentration of 0.018% w/v, which is higher that lung exposure estimated from the amount of liposomes providing suitable SC pulmonary dose.

In the revised version the explanation has been extended as follows:

“After 24 h the culture medium was removed and substituted by fresh culture medium mixed with sterile liposome suspension with a dilution of 1:100, that corresponds to 0.018% w/v. This concentration is higher than lung exposure estimated from the amount of liposomes providing suitable SC pulmonary dose”. The mixture was maintained ….(lines 171-173).

Higher magnification of images have been included in the revised version (600 dpi instead of 300 dpi).

With respect to the SEM images, differences in liposome size are only relevant for fresh liposomes (266.60±2.08 and 272.41±7.00 for lactose ans sucrose, respectively after filtration though 0.45 µm versus 205.40±11.40 and 190.51±17.44 after flitration through 0.22 µm). Such differences can be appreciated in the SEM figure (upper and lower left-side pictures in figure 5). For rehydrated liposomes, however, differences between 0,45 and 0.20 µm are not relevant, neither in the table 3 (Dh values) nor in the figure.

  • To better highlight the advantages of the additive it would be nice to include also the characterization of liposomes without additives after lyophilization. Changes in biocompatibility, size, PDI and drying rate should be reported and compared with the ones of liposomes with additives.

There is wide literature information that confirm drug leaking and vesicle aggregation/fusion when liposomes are lyophilized without lyoprotectant additives. Moreover, research is being done on this topic and results reveal that it is difficult to find the conditions for liposome stabilization and that each formulation has to be carefully optimized. According to our opinion, the performance of all experimets with liposome formulations without additive will not provide further knowledge on this topic.

Reviewer 2 Report

Aim of this study is the evaluation of the lyoprotective effects of mannitol and lactose among sildenafil citrate liposomes. Liposomes were characterized before and after the lyophilization in term of size, entrapment efficiency, thermal properties, biocompatibility and cell internalization . I was found that the type of additive did not affect the biocompatibility and cell internalization but did influence other liposome attributes. Lactose resuts the best choice for the development of sildenafil citrate liposomes.

The paper is quite well written and the results of this research are interesting and fit in the Pharmaceutics field. 

Just two comments in order to improve the paper clearness:

The Author use citrate buffer pH 3.2, Are they sure that this value is not irritant for the pulmunar tissues?

Taking into account that this formulation is aimed for pulmonar delivery can the Authors provide some information about the aerodynamic properties of the system?

Author Response

REVIEWER 2

Comments and Suggestions for Authors

Aim of this study is the evaluation of the lyoprotective effects of mannitol and lactose among sildenafil citrate liposomes. Liposomes were characterized before and after the lyophilization in term of size, entrapment efficiency, thermal properties, biocompatibility and cell internalization. I was found that the type of additive did not affect the biocompatibility and cell internalization but did influence other liposome attributes. Lactose resuts the best choice for the development of sildenafil citrate liposomes.

The paper is quite well written and the results of this research are interesting and fit in the Pharmaceutics field.

Just two comments in order to improve the paper clearness:

We thank the reviewer comments and we agree with the reviewer on the  interest of  suggested studies

The Author use citrate buffer pH 3.2, Are they sure that this value is not irritant for the pulmunar tissues?

Answer:  citrate buffer pH= 3.2 was used to prepare the mixture, then the pH was adjusted to 7.0 (explained in section 2.2.1), therefore, the pH of the final formulation was 7.0

Taking into account that this formulation is aimed for pulmonar delivery can the Authors provide some information about the aerodynamic properties of the system

Answer: this is a relevant issue and it is planned to do studies on this topic in the near future. The aim of the present work was to evaluate the lyoprotective effects of lactose and manitol compared to estandard lyoprotectants (sucrose and trehalose) and to determine the conditions for liposome stabilization by lyophilization. We checked that the liposome size after lyophilisation and rehydration was suiable por pulmonary drug delivery.  The aerodynamic properties depend on many factors including the type of device used for pulmonary administration, such as nebulizers in case the liposomes are delivered as rehydrated product or dry powder inhalers in caes the liposomes are delivered as the freeze-dried powder. This will be the next step of the study.

Reviewer 3 Report

I have carefully read the paper of de Jesús Valle et al. about the lyophilization of liposomes, and in particular on the role of some additives (bulk agent, lyoprotectant, cryoprotectant). The paper is interesting but requires a revision regarding the lyophilization section.

  1. The authors have performed annealing at -33 °C, but they didn't measure the glass transition temperature of the frozen formulations (Tg'). Since annealing has to be performed in a range of temperature between Tg' and ice melting temperature, such information is essential. Moreover, the authors did not indicate the duration of the annealing. Please provide Tg' for every formulation and all the cycle variables.
  2. Can the authors double-check if the chamber pressure of 0.008 mBar is the correct pressure?
  3. line 302: "Table 2 shows the influence of annealing on primary drying". Actually, the authors have studied the influence of annealing on the residual moisture (at the equilibrium) and not on primary drying. In fact, annealing can influence the primary drying only in the sense of a reduction of primary drying duration. Other considerations have to be taken cautiously because of the many variables involved.
  4. In my opinion, the lyophilization section cannot be discussed as the authors did because the equipment used can not allow the control of many fundamental parameters:
      1.  Freezing step: cooling rate? The cooling rate has a strong connection with the primary drying rate and secondary drying rate because it influences the size and the shape of ice crystals. Usually, faster freezing implies smaller ice crystals. On the other hand, annealing increases the size of ice crystals.
      2. Primary drying: product temperature and primary drying duration. The product temperature is usually measured using thermocouples and primary drying duration using Pirani/Baratron pressure ratio. If the cycle is stopped before all the free water is sublimated, an increase in temperature leads to melting.
      3. Secondary drying: product temperature and secondary drying duration. Secondary drying duration is established on the base of the desired final moisture. 

The comments and conclusions of the authors regarding the annealing are statistically meaningless without the control of the above parameters. In fact:

a. The effect of annealing on primary drying has to be seen in terms of drying rate and not in terms of residual moisture after a long primary drying duration (24 h).

b. The effect of annealing on secondary drying has to be seen in terms of desorption rate and not in terms of final residual moisture after a long secondary drying (20 h)

In both cases, what the authors have measured is simply the residual moisture at the equilibrium (at the temperature and pressure set by the authors). That is probably why they didn't find statistical differences between products with and without annealing. Moreover, the effectiveness of annealing depends on the formulation (see the comment above)

5 - Annealing is able to increase the ice crystals size and, so, the pores of the cake. Did you notice differences in terms of EE in the case of freeze-drying with and without annealing?

typos:

line 194: four percent -> 4 %at 

Author Response

REVIEWER 3

Comments and Suggestions for Authors

I have carefully read the paper of de Jesús Valle et al. about the lyophilization of liposomes, and in particular on the role of some additives (bulk agent, lyoprotectant, cryoprotectant). The paper is interesting but requires a revision regarding the lyophilization section.

We appreciate very much the reviewer´s comments and suggestions and we have made the corresponding changes in the revised version

The authors have performed annealing at -33 °C, but they didn't measure the glass transition temperature of the frozen formulations (Tg'). Since annealing has to be performed in a range of temperature between Tg' and ice melting temperature, such information is essential. Moreover, the authors did not indicate the duration of the annealing. Please provide Tg' for every formulation and all the cycle variables.

Answer: We agree with the reviewer about the interest of measuring the Tg´ of the frozen formulations. Unfortunately, the available DSC equipment did not allow scanning from very low temperatures.  Instead, we considered the literature data about eutectic (Te) and transition temperatures reported for the additives used: manitol, Te = -2º C. Lactose,               T´g = -28ºC. Sucrose, T´g= -30º C. Trehalose, T´g = -28ºC (Chen C et al., 2010. J. Control Release). Since annealing did not improve results and this was discarded, we think that the experimental measure of Tg´ may not be essential in our study.

With respect to the duration of the annealing, it was 4 hours. This information has been included in the revised versión of the ms. (line 201)

Can the authors double-check if the chamber pressure of 0.008 mBar is the correct pressure?

Answer: We confirm this value. We know that this is a very low pressure but the equipment used (Telstar Cryodos) displays the chamber pressure and, after 15- 20 min all values were in the range of 0.008±0.002 mBar

line 302: "Table 2 shows the influence of annealing on primary drying". Actually, the authors have studied the influence of annealing on the residual moisture (at the equilibrium) and not on primary drying. In fact, annealing can influence the primary drying only in the sense of a reduction of primary drying duration. Other considerations have to be taken cautiously because of the many variables involved.

Answer: We agree with the reviewer, in fact we studied the influence of annealing on the residual moisture (RM). Since RM depends on drying rate, indirectly we studied the influence of annealing on primary drying, but we agree that is more correct to say that “the influence of annealing on remaining moisture” was studied. This has ben changed in the text (line 309).

In my opinion, the lyophilization section cannot be discussed as the authors did because the equipment used can not allow the control of many fundamental parameters:

Freezing step: cooling rate? The cooling rate has a strong connection with the primary drying rate and secondary drying rate because it influences the size and the shape of ice crystals. Usually, faster freezing implies smaller ice crystals. On the other hand, annealing increases the size of ice crystals.

Answer: In fact, the equipment used did not allow selection of parameters, such as cooling rate, but this guarantied the same conditions for all samples.  Since all samples underwent the same cycle conditions, the differences in RM among samples are only imputed to the additive used that may influence the size and shape of crystals and, finally, the driying rate.   Our study was focused on evaluating the lyoprotective effects of manitol and lactose compared to sucrose and trehalose. Despite the technical limitations of the equipment used, we still think that the result obtained are very valuable with regard to the potential use of lactose as lyoprotectant for liposomal formulations aimed at pulmonary drug delivery. In particular, for SC liposomes prepared as described in the method section.

Primary drying: product temperature and primary drying duration. The product temperature is usually measured using thermocouples and primary drying duration using Pirani/Baratron pressure ratio. If the cycle is stopped before all the free water is sublimated, an increase in temperature leads to melting.

Answer: Unfortunately, our equipment is not provided with either thermocouples or product RM probes. Instead, we did meassure the RM of samples after different periods of drying time and we observed the aspect of such samples at room temperature. Since all the frozen (free) water is removed during primary drying, there is not sample collapse at room temperatura as far as primary drying is completed. On the contrary, if primary drying had not been completed, the frozen  water melt at room temperature and samples immediately collapse.  We observed that samples with remaining moisture under 5% did not collapse at room temperature,  while those with RM ≥6% immediately did melt-collapsed at room temperature. According to this, we hypothesized that this value corresponds to the unfrozen (bound) water in our samples. Therefore 5% RM is the cut-off value for safely removing unfrozed water by desorption during secondary drying.  According to our opinion, this is an interesting finding.  As the reviewer has commented many lyophilizers, in particular those used for industrial production, are provided with several control systems including RM product control.  If the RM corresponding to the unfrozen water is known for a particular formulation, this can be used as reference for starting secondary drying, irrespectively of the type of lyophilizer used.

Secondary drying: product temperature and secondary drying duration. Secondary drying duration is established on the base of the desired final moisture.

Answer: Secondary drying was first carried out at 10ºC and 15ºC, but very slow reduction of RM was observed. Then a final temperatura of 20ºC was selected (infomation included in table 3 and commented in 3.3 Section, lines 329-322).

Since optimal RM for liposomes is not reported,  several periods of secondary drying were assayed and the RM and Tg of samples were determined. A duration of 24 h was selected since this lead to suitable RM and Tg values for lactose and sucrose.

The comments and conclusions of the authors regarding the annealing are statistically meaningless without the control of the above parameters. In fact:

a.The effect of annealing on primary drying has to be seen in terms of drying rate and not in terms of residual moisture after a long primary drying duration (24 h).

  1. The effect of annealing on secondary drying has to be seen in terms of desorption rate and not in terms of final residual moisture after a long secondary drying (20 h)

In both cases, what the authors have measured is simply the residual moisture at the equilibrium (at the temperature and pressure set by the authors). That is probably why they didn't find statistical differences between products with and without annealing. Moreover, the effectiveness of annealing depends on the formulation (see the comment above

Answer: The annealing was not assayed for samples that underwent secondary drying, but just for samples dryied for 24 h at temperature under Tg´value. Table 2 is the only one showing results from assays with annealing. The rest of experiments were carried out without annealig. This is commented in section 3.3, lines 322-323 (“Since annealing did not improve the primary drying, the rest of experiments were performed without annealing)

5 - Annealing is able to increase the ice crystals size and, so, the pores of the cake. Did you notice differences in terms of EE in the case of freeze-drying with and without annealing?

Answer: Right, this likely happened but we did not check this. As previously commented, we compared the RM of samples after 24 h of primary drying and we found that annealing did not improve results. Therefore, we discarded annealing and made the rest of experiments without annealing.

typos: line 194: four percent -> 4 %at

Answer: four percent has been substituted by 4% in the revised version (line 198).

Reviewer 4 Report

The paper “Lyoprotective effects of mannitol and lactose compared to sucrose and trehalose. Sildenafil citrate liposomes, as a case study” by María José de Jesús Valle at al. deals with the preparation of sildenafil citrate-loaded liposomes for pulmonary delivery. Different possible excipients (mannitol, lactose, sucrose and trehalose) are tested, and the effect of lyophilization is also investigated. The subject is interesting, and the paper is overall well-written.

However, some points need to be addressed before this manuscript may be published:

Abstract:

“A cut-off of 5% […]” please specify if it is a w/w value, maybe on a dry product basis?

Materials and Methods:

“Sodium Hydroxide (NaHCO3) […]” was it sodium hydroxyde or sodium hydrogen carbonate?

“Fetal Bovine Serum (FBS) and Bovine Serum Albumin (BSA) was purchased […]” was should be were

Lines 149 and 161: “The instrument was calibrated with Milli-Q water (ρ = 1g/ml; η = 1 mm2/s) and the viscometer constant (K, mm2/s2) was estimated.” “[…] flasks of 75 cm2 [...]Please check the use of superscripts. Overall, please check the use of superscripts/subscripts throughout the whole text, for instance in the cellular density (2 x 104 cells/well in line 168, 4x104 cells/well in line 184).

Section 2.4: Many details are missing. What was the annealing time at -33 °C? Was the sampled cooled back again after annealing, and before starting primary drying? In that case, how was cooling performed? What was the shelf temperature during primary drying? Are the authors sure that pressure was 0.008 mbar? This value is extremely low and hard to reach by most equipment. How was the cycle monitored? Did the authors used thermocouples or other temperature sensors for the product? How was pressure monitored/controlled (MKS Baratron or Pirani sensors or both)? How was the end of primary drying determined? How long was the ramp between the primary and secondary drying temperatures? Was secondary drying performed at the same chamber pressure as primary drying?

Line 212: “The samples (1.0 mg) […]” such sample mass seems very small for DSC analyses. What type of pan was used?

Line 213-214: what was the cooling rate to -20°C used in the DSC runs?

Results:

Figure 1: please use the point as decimal separator on the y-axis.

Tables 1, 2 and 3: you could show the results of the ANOVA analysis here, for instance marking statistically different values with different letters.

Figures 2, 3, 4: why didn’t the authors use mannitol or trehalose for these analyses?

Table 2: Please specify what the moisture content values herein listed refer to. Are they w/w percentages, on a dry product basis?

Table 3 caption: “and zeta potential or rehydrated liposomes” or should be of

Lines 334-336: “The scanning electron microscopy (SEM) images of lyophilized cakes (Figure 5) revealed that the products had an amorphous morphology that is beneficial to the stability of the samples.” How can the authors distinguish the amorphous state of a lyophilized cake by SEM inspection? XRD analysis would be needed for this purpose. I am actually pretty sure that, for instance, mannitol crystallized during freeze drying in their formulations (at least partially). Such crystallization explains the lower RM they observed in the SC+M formulation. For this reason, I believe the authors cannot claim their products were amorphous unless an XRD investigation is performed.

Figure 5: what was the additive present in the samples analyzed by SEM and showed in the figure? The authors could show SEM images of all the different formulations tested in their work. Also, please make the scale bar clearer (I guess it corresponds to 2 um, but the legend under the figures is a bit blurry).

Figure 6: please use the point as decimal separator. Why was this analysis performed only for lactose and sucrose?

Table S1, Figure 7: The results for trehalose look weird. How can the Tg value increase when increasing the moisture content? Water should act as a plasticizer. Also, the Tg should appear in the DSC thermograms as an inflection point, as the Tg is a change in specific heat capacity, without any enthalpic gain/loss. What the authors are showing looks instead like an endothermic peak (i.e., en enthalpic transition, like a melting). Are they sure they are really monitoring a Tg?

Discussion:

Lines 425-426: “An interesting finding in this study was the determination of a RM cut-off value of 5% that would allow for safe secondary drying.” Actually, collapse is observed whenever the product temperature overcomes a product-specific value, which is related to the glass transition. In turn, the glass transition temperature is influenced by the moisture content (as observed by the authors), because water acts as a plasticizer. It may be that, during secondary drying, the product temperature overcomes the collapse value of the formulation, if the residual moisture in the product is too high and the glass transition value is too low. However, this consideration is extremely product-specific (because the thermal behavior strongly depends on the formulation) and I would recommend the authors to be less conclusive in their statement, as it may not apply to different systems.

Please check the spelling of trehalose throughout the text.

Author Response

REVIEWER 4

Comments and Suggestions for Authors

The paper “Lyoprotective effects of mannitol and lactose compared to sucrose and trehalose. Sildenafil citrate liposomes, as a case study” by María José de Jesús Valle at al. deals with the preparation of sildenafil citrate-loaded liposomes for pulmonary delivery. Different possible excipients (mannitol, lactose, sucrose and trehalose) are tested, and the effect of lyophilization is also investigated. The subject is interesting, and the paper is overall well-written.

However, some points need to be addressed before this manuscript may be published:

We thank all corrections and suggestions from the reviewer. We have considered all of them and we have made the corresponding chages in the manuscript

Abstract:

“A cut-off of 5% […]” please specify if it is a w/w value, maybe on a dry product basis?

Answer:  yes, this is a w/w value on the dry product basis. This has been explained in the revised version (line 25 and line 211).

Materials and Methods:

“Sodium Hydroxide (NaHCO3) […]” was it sodium hydroxyde or sodium hydrogen carbonate?

Answer: It was sodium hydroxide (NaOH). This has been s corrected in the revised version line 93.

“Fetal Bovine Serum (FBS) and Bovine Serum Albumin (BSA) was purchased […]” was should be were

Answer: Right, it was a mistake. This is changed in the revised versión line 95.

Lines 149 and 161: “The instrument was calibrated with Milli-Q water (ρ = 1g/ml; η = 1 mm2/s) and the viscometer constant (K, mm2/s2) was estimated.” “[…] flasks of 75 cm2 [...]” Please check the use of superscripts. Overall, please check the use of superscripts/subscripts throughout the whole text, for instance in the cellular density (2 x 104 cells/well in line 168, 4x104 cells/well in line 184).

Answer: The superscripts and subscripts have been revised and corrected all over the text

Section 2.4: Many details are missing. What was the annealing time at -33 °C? Was the sampled cooled back again after annealing, and before starting primary drying? In that case, how was cooling performed?  What was the shelf temperature during primary drying? Are the authors sure that pressure was 0.008 mbar? This value is extremely low and hard to reach by most equipment. How was the cycle monitored? Did the authors used thermocouples or other temperature sensors for the product? How was pressure monitored/controlled (MKS Baratron or Pirani sensors or both)? How was the end of primary drying determined? How long was the ramp between the primary and secondary drying temperatures? Was secondary drying performed at the same chamber pressure as primary drying?

Answer: The annealing time was 4 h and the samples were not cooled back after annealing since the Te and Tg values of additives (manitol, Te = -2º C; lactose, T´g = -28ºC; sucrose, T´g= -30º C; trehalose, T´g = -28ºC (Chen C.  et al.,  J Control Release, 2010) are above annealing temperature. Duration of anneling has been included in the revised ms (line 200).

We confirm the chamber pressure value. We know that this is a very low pressure but the equipment used (Telstar Cryodos) continually displays the chamber pressure and all displayed values were in the range of 0.008±0.002 mBar after 15-20 from starting drying.

Unfortunately, our equipment is not provided with thermocouples, therefore the product temperature was not monitored. The chamber pressure was monitored with Pirani sensor.

With respect to secondary drying, we did meassure the RM of samples after different periods of drying time and we observed the aspect of such samples at room temperature. We observed that samples with remaining moisture under 5% did not collapse at room temperature while those with RM ≥6% immediately did melt-collapsed at room temperature. Since all the frozen water is removed during primary drying, there is not sample collapse at room temperature as far as primary drying is completed. On the contrary, if primary drying had not been completed, the frozen water melt at room temperature, and then samples immediately collapse.  According to this, we hypothesized that this value corresponds to the unfrozen water for our formulation and 5% RM was the reference for starting the secondary drying. Therefore, the end of primary drying was when samples achive RM ≤ 5% w/w, on dried product basis. After several assay-error trials, 24 h of primary drying was found to be the time required by our equipment.

Line 212: “The samples (1.0 mg) […]” such sample mass seems very small for DSC analyses. What type of pan was used?

40 µL pans were used. This information   has been included in the revised ms. (line 217)

Line 213-214: what was the cooling rate to -20°C used in the DSC runs?

Answer: 10ºC/min. This information is in 2.6 section, line 218

Results:

Figure 1: please use the point as decimal separator on the y-axis.

Answer: Right, it was a mistake and has been corrected in the revised version

Tables 1, 2 and 3: you could show the results of the ANOVA analysis here, for instance marking statistically different values with different letters.

Answer: one-way ANOVA was performed to compare

In table 1

  • Samples without additive with those that contain additive,
  • Liposomes filtrered though 0.45 µm with those filtered through 0,20 µm

In table 2

  • Samples with annealing and samples without annealing
  • Liposomes filtrered though 0.45 µm with those filtered through 0,20 µm

In table 3

  1. Fresh liposomes with rehydrated liposomes.
  2. Liposomes filtrered though 0.45 µm with those filtered through 0,20 µm

Since each table shows results of two comparisons, the inclusion of letters would be confuse. We think that commentaries in the text on the existence of statitistical significance, or not, is more clear.

Figures 2, 3, 4: why didn’t the authors use mannitol or trehalose for these analyses?

Answer: Since mannitol and trehalose showed worse results we did not performed this anlysis with those samples

Table 2: Please specify what the moisture content values herein listed refer to. Are they w/w percentages, on a dry product basis?

Answer: Yes, the values are % w/w on a dry product basis. This has been clarified in table 2 caption  (line 316).

Table 3 caption: “and zeta potential or rehydrated liposomes” or should be of

Answer: Yes, it was mistake it has been corrected (line 345).

Lines 334-336: “The scanning electron microscopy (SEM) images of lyophilized cakes (Figure 5) revealed that the products had an amorphous morphology that is beneficial to the stability of the samples.” How can the authors distinguish the amorphous state of a lyophilized cake by SEM inspection? XRD analysis would be needed for this purpose. I am actually pretty sure that, for instance, mannitol crystallized during freeze drying in their formulations (at least partially). Such crystallization explains the lower RM they observed in the SC+M formulation. For this reason, I believe the authors cannot claim their products were amorphous unless an XRD investigation is performed.

Answer: We agree with the reviewer, XRD analysis is needed for identifiying the crystal structure of samples. SEM images only provides the surface morphology of samples. XRD analysis was not performed. SEM analysis was performed with lactose samples (explained in the figure caption of the revised ms,  line 349). For this, SEM images revealed a smooth surface withouth typical cystal edges. For this reason we though that SEM confirmed the amorphous morphology of samples, but according with the reviewer comment this has beeh changed in the revised version and the sentence “The scanning electron microscopy (SEM) images of lyophilized cakes (Figure 5) revealed that the products had an amorphous morphology” has been substituted by “The scanning electron microscopy (SEM) images of lyophilized cakes (Figure 5) revealed a smooth surface and no evidence of crystal edges”(lines 339-340).

Figure 5: what was the additive present in the samples analyzed by SEM and showed in the figure? The authors could show SEM images of all the different formulations tested in their work. Also, please make the scale bar clearer (I guess it corresponds to 2 µm, but the legend under the figures is a bit blurry).

Answer: The additive was lactose. Since lactose was the additive of interest for pulmonary delivery this analysis was performed with lactose. 

In fact, the scale bar is 2 µm and the legend under the figure 5 was blurry. Higher magnification images have been included in the revised version (600 dpi instead of 300 dpi).

Figure 6: please use the point as decimal separator. Why was this analysis performed only for lactose and sucrose?

Answer: allright, the decimal separator has been changed in the revised ms

This analysis was done with lactose and sucrose because these two additives showed better lyoprotective  results

Table S1, Figure 7: The results for trehalose look weird. How can the Tg value increase when increasing the moisture content? Water should act as a plasticizer. Also, the Tg should appear in the DSC thermograms as an inflection point, as the Tg is a change in specific heat capacity, without any enthalpic gain/loss. What the authors are showing looks instead like an endothermic peak (i.e., en enthalpic transition, like a melting). Are they sure they are really monitoring a Tg?

Answer: yes, we agree with the reviewer that the Tg results for trehalose look weird. Interpretation of thermograms for trehalose was not easy.  Experiments were repeated but the graphs were very smooth and the onset was not clear.  Therefore, the Tg values were not accurately estimated for trehalose. Since the work was focused on lactose we did not investigate further for trehalose. For the rest of additives the Tg was monitored and estimated.

Discussion:

Lines 425-426: “An interesting finding in this study was the determination of a RM cut-off value of 5% that would allow for safe secondary drying.” Actually, collapse is observed whenever the product temperature overcomes a product-specific value, which is related to the glass transition. In turn, the glass transition temperature is influenced by the moisture content (as observed by the authors), because water acts as a plasticizer. It may be that, during secondary drying, the product temperature overcomes the collapse value of the formulation, if the residual moisture in the product is too high and the glass transition value is too low. However, this consideration is extremely product-specific (because the thermal behavior strongly depends on the formulation) and I would recommend the authors to be less conclusive in their statement, as it may not apply to different systems.

Answer: Yes, we agree, this consideration is product-specific. This applies only to the formulations tested in this work. This has been carified in the revised version: The paragraph “This finding is more useful for scaling up the procedure in the transition from laboratory to large-scale production than technical information on the conditions and/or characteristics of the equipment used” has been substituted by: “ This value is specific for the formulations tested here and facilitates the scaling up in any lyophilization equipmen,t as far as the remaining moisture is monitored”  (lines 433-435).

Regarding the finding of 5% as a starting point for secondary drying, we have addressed this issue in previous answer. We did meassure the RM of samples after different drying time and we observed the aspect of such samples at room temperature. It was observed that samples with remaining moisture under 5% did not collapse at room temperature while those with RM ≥6% immediately did melt-collapsed at room temperature. Since all the frozen water is removed during primary drying, there is not sample collapse at room temperatura as far as primary drying is completed. On the contrary, if primary drying had not been completed, the frozen water melt at room temperature, and then samples immediately collapse.  According to this, we hypothesized that this value corresponds to the unfrozen (bound) water and 5% RM can be used as the reference for starting the secondary drying, irrespectively of the lyophilizer used. This applies to our formulation and provides relevant information for scale-up to any type of equipment

Please check the spelling of trehalose throughout the text.

Answer: OK, this has been done and corrected in the figure 7.

Round 2

Reviewer 4 Report

The paper “Lyoprotective effects of mannitol and lactose compared to sucrose and trehalose. Sildenafil citrate liposomes, as a case study” by María José de Jesús Valle at al. improved after the first round of revision. However, the following minor points should still be addressed before publication.

Section 2.4: The annealing step is only useful if conducted above the glass transition value of the formulation, as only in these conditions the soild-phase modifications that should be promoted during annealing can occur at a reasonable rate. The authors should explain in larger detail their choice for the annealing step. Also, the value of the shelf temperature during primary drying should be indicated. Was secondary drying performed at the same chamber pressure as primary drying? Please indicate that a Pirani gauge was used for measuring pressure.

Table S1, Figure 7: I would recommend adding a comment to the manuscript to explain that the Tg values for trehalose could not be accurately estimated.

Author Response

This manuscript is a resubmission of an earlier submission. The following is a list of the peer review reports and author responses from that submission.